# Manganese Pollution and Its Remediation: A Review of Biological Removal and Promising Combination Strategies

**DOI:** 10.3390/microorganisms10122411

**Published:** 2022-12-06

**Authors:** Rongrong Wu, Fangting Yao, Xiaoya Li, Chongjing Shi, Xue Zang, Xiao Shu, Hengwei Liu, Wenchao Zhang

**Affiliations:** School of Chemistry and Life Sciences, Suzhou University of Science and Technology, Suzhou 215009, China

**Keywords:** removing manganese, manganese-oxidizing bacteria, biosorption, bioaccumulation, heavy metal, bio-oxidation, microbially induced carbonate precipitation (MICP)

## Abstract

Manganese (Mn), as a cofactor of multiple enzymes, exhibits great significance to the human body, plants and animals. It is also a critical raw material and alloying element. However, extensive employment for industrial purposes leads to its excessive emission into the environment and turns into a significant threat to the ecosystem and public health. This review firstly introduces the essentiality, toxicity and regulation of Mn. Several traditional physicochemical methods and their problems are briefly discussed as well. Biological remediation, especially microorganism-mediated strategies, is a potential alternative for remediating Mn-polluted environments in a cost-efficient and eco-friendly manner. Among them, microbially induced carbonate precipitation (MICP), biosorption, bioaccumulation, bio-oxidation are discussed in detail, including their mechanisms, pivotal influencing factors along with strengths and limitations. In order to promote bioremediation efficiency, the combination of different techniques is preferable, and their research progress is also summarized. Finally, we propose the future directions of Mn bioremediation by microbes. Conclusively, this review provides a scientific basis for the microbial remediation performance for Mn pollution and guides the development of a comprehensive competent strategy towards practical Mn remediation.

## 1. Introduction

The rapid development of industrialization and human activities lead to more and more serious heavy metal pollution in the world. It has aroused widespread concern. Manganese (Mn), a naturally occurring element, is commonly found in soil, water and rocks. As a significant trace element for the human body, Mn is indispensable for regulating individual metabolism, digestion, reproduction, defending antioxidants and neuronal health [1,2,3]. For plants, Mn also plays a role in diverse processes of a plant’s life cycle such as photosynthesis [4], respiration, scavenging of reactive oxygen species (ROS) [5], pathogen defense [6], and hormone signaling. Similarly, Mn functions uniquely and ubiquitously in regulation high ROS levels in bacteria, because during the process Fenton reaction will not be activated [7].

Mn has been widely used in production of non-ferrous metallurgy, steel, batteries, electrode materials and catalyst [8]. In production, metal Mn is a superior kind of alloy element. Thus, Mn is extensively employed in certain types of steel, particularly low-carbon steels (e.g., 200 series) and nonferrous alloys [9]. The stacking of electrolytic manganese residue (EMR) leads to the infiltration of leachate and has become the largest source of Mn contamination in soil [10]. Although Mn is considered as a basic and essential nutrient for living organisms, overexposure of Mn may result in hazardous and irreversible damage to the ecosystem and human health. For example, excessive Mn may influence the respiration and metabolism of the microorganisms in soil and cause a decrease of organic carbon [11]. Physiological dysfunction and malnutrition of plants also happen in this case. Additionally, accumulation of Mn in excess in the central nervous system (CNS) can bring about a disease named as manganism (detailed information in Section 2.3) which resembles Idiopathic Parkinson’s Disease (IPD) clinically and results in adverse neurological effects in human [12]. As a kind of heavy metal, Mn is difficult to be metabolized naturally and will gradually migrate in environment and accumulate in living beings through the food chain, which is hard to control. Thus, the treatment of Mn contamination admits of no delay [13].

Common methods of treating Mn pollution primarily consist of physical (such as ion exchange method and adsorption), chemical, biological and other novel techniques including electrochemistry and flocculation method [14,15]. Specifically, hydroxide precipitation and carbonate precipitation are discussed extensively, for the former eliminates Mn element selectively and happens in multi-stage, while the latter exhibits enormous economic benefits. In addition, oxidation, as a category of chemical treatment, is studied preferentially as well. Because it is able to remove Mn in the range of low pH values with the aid of strong oxidizers. Nevertheless, most physical and chemical treatments are expensive, complicated and hard to assess their long-term effectiveness [11]. What is worse, since the accuracy of real-time Mn concentration in effluents is difficult to ensure, there is no guarantee that chemicals that are dumped for proceeding chemical treatment be completely consumed in the process, so that waste and secondary pollution occur. Therefore, biological treatment distinguishes itself for that it is not only cost-efficient, but it requires no chemical oxidants, causes no toxic by-products and the bacteria can be enriched rapidly. 

This article mainly reviews some categories of biological methods for removing Mn through microorganisms, including biosorption, bioaccumulation, biological oxidation, microbially induced carbonate precipitation (MICP), from their mechanisms, key factors to advantages and limitations. The biologically−based combination approaches and other methods are summarized comprehensively. Finally, four needed directions are proposed for advancing the Mn bioremediation research. In summary, this review provides an in-depth analysis of biological removal of Mn by microorganisms, which may help to assess and guide the selection of optical heavy metal remediation techniques for improvement in actual applications.

## 2. Mn in the Environment

### 2.1. Mn Characteristics and Essentiality

Mn is a kind of grayish-red transition metal element, which is one of the most abundant elements in the earth’s crust with a proportion of 0.095%. It exists in a variety of ores, oxides, carbonates, and it is widely distributed in soil, water and sediments. Generally, Mn has valence states including +2, +3, +4, +6 and +7, and mainly remains in the form of Mn^2+^ in waters at pH 4–7, and higher oxidation forms will be present at higher pH or due to microbial oxidation [16].

Mn exhibits beneficial effects chiefly in terms of incorporating metal into metalloproteins. Additionally, it is incorporated into arginase, glutamine synthetase, phosphoenolpyruvate, decarboxylase, pyruvate carboxylase, and Mn superoxide dismutase enzymes [2]. Taking arginase as an example, maintaining ammonia levels in the body relies on Mn to control the activity of arginase. Additionally, bone health is greatly influenced by Mn as it is capable of combining with glycosyltransferases [17]. A recent study discovered that in the perspective of self-defense, not only does Mn promote a bodies’ ability to defend DNA virus by increasing the sensibility of the DNA senor cGAS and its downstream adaptor protein STING, but it plays an important role in inherent immune sensing of tumors and enhances adaptive immune responses against tumors [18].

Furthermore, quite a few research have suggested that Mn bioavailability could be considered as a major regulator of long-term litter decomposition rates [19]. The fact that Mn is an essential element for the production of Mn peroxidase (MnP), an important enzyme for the breakdown of lignin and humification products, probably accounts for the phenomenon [20].

### 2.2. Source and Geochemical Cycle of Mn

In nature, Mn is mainly derived from deep-sea Mn ore clusters, rock crust, soil and freshwater water. Crustal rocks and mining ores formed by volcanic eruption, forest fire and various vivid plants on land and water are the main sources of Mn reserves in the biosphere [21]. The reserves of Mn are concentrated in the developing countries such as Australia, Russia, India, Brazil and South Africa. 

The primary source of Mn intake in non-occupational settings is through the diet, with adults consuming 1–10 mg Mn/day, approximately 1–5% of which is absorbed in the gut [22]. Mn is quite abundant in wheat germs, green vegetables, tea, highly refined bread and cereals rooted in fertile soil. Additionally, Mn levels are high in almost all dry fruits, especially pine nuts [23,24].

At present, China is the largest producer, consumer, and exporter of electrolytic Mn products in the world [25]. Although Mn metal has greatly contributed to industrial development and regional economic construction, it has also given rise to serious environmental pollution and damage, the most worrying of which is the pollution caused by EMR [26,27,28,29]. EMR is a type of waste residue generated during the production of electrolytic Mn after leaching with concentrated sulfuric acid, neutralization with ammonia, and filtration by plate filter pressing [30,31]. Many waste rocks are produced in the process of continuous mining in the Mn ore area, resulting in the formation of rock storage yard. Under the effect of rainfall leaching, the Mn-contained leachate from waste rock storage yard migrates with the rainfall infiltration and surface runoff and enters into the soil, surface, and underground water of the mining area, resulting in Mn pollution of the ecological system [32,33]. In addition, the stacking sites of EMR are often located in open sites near the plants, and these sites are usually not well managed to prevent toxic substances from releasing into the soil and surrounding areas. This not only occupies massive land resources but causes serious pollution of surrounding soil and receiving water bodies [34]. Plants grown from soil, which is contaminated with heavy metals or irrigated with contaminated water will find severe accumulation of heavy metals in their bodies, and then influence the food chain down to animals and human beings. 

The main sources of airborne Mn are industrial emissions, such as foundries and ferro-Mn facilities, combustion of fossil fuels, and entrainment of Mn-containing soils [35]. Those people who accidentally inhale Mn may experience that Mn does not pass through the liver but can be directly transported into the brain by olfactory or trigeminal presynaptic nerve ending transport. In the brain, Mn disrupts dopamine, serotonin, and glutamine signaling [36,37] and people are likely to suffer manganism [38,39]. It deserves to be noted that although methylcyclopentadienyl Mn tricarbonyl (MMT), which has been used in leaded gasoline, unleaded gasoline, diesel fuel, fuel oil, and turbine fuel to raise octane and improve combustion, contains approximately 24.4% Mn and has a potential to be released into the air, the soil along busy roads is not elevated in Mn, even after long-term use of MMT [40]. Some studies have found that seasonal temperature stratification is the main trigger of summer reservoir Mn pollution [41], which is also verified by a reservoir Mn pollution model [42]. For production and daily life, water with excessive Mn is undesirable for customers due to discoloration of the water and the subsequent staining of laundry and plumbing fixtures [43], as well as a strong corrosion capacity to the production equipment [32]. Therefore, harmless disposal of Mn pollution now admits of no delay. The process of Mn contamination generation and migration in ecological environment are illustrated in Figure 1.

### 2.3. Mn Toxicity

For the human body, pollution from heavy metals is of great concern as they are non-biodegradable and are accumulated easily in the food chain [44]. According to the water quality standard of the World Health Organization (WHO), the content Mn in drinking water should not exceed 0.1 mg/L. Sidoryk-Wegrzynowicz et al. assumed that excessive accumulation of Mn in the CNS triggers neurotoxicity, which results in a neurological brain disorder, referred to as manganism [45]. In the early stages of the disease, patients display psychotic symptoms, which gradually develops into chronic disturbances in extrapyramidal circuits, leading to postural instability, dystonia and bradyskinesia, micrographia, mask-like facial expression, and speech disturbance [46,47]. The effect of Mn on the brain is also influenced by route of exposure and magnitude of accumulation. Inhalation of Mn is generally associated with oxidative stress and increased neuronal apoptosis, whereas ingestion of Mn has more of a subtle effect, altering neurochemistry and cognition. At present, Mn toxicity is mainly associated with occupational exposure of welders, miners, and steel workers to chronic high levels of airborne particulate Mn. Once inhaled, Mn can lead to inflammation in the lungs and respiratory symptoms gradually, including cough, bronchitis, pneumonitis, and impaired pulmonary function [17]. Mn smelters and miners have a propensity to develop both occupational manganism and amyotrophic lateral sclerosis (ALS).

For animals, inhalation of particulate Mn compounds (e.g., Mn dioxide [MnO_2_] or Mn tetroxide [Mn_3_O_4_]) leads to an inflammatory response in their lungs, and Mn-induced neurotoxicity is considered to be one of the most sensitive toxicological endpoints [48]. As far as specific reports, it was found that high-levels of Mn accumulating in certain parts of a quolls’ body could influence the viability of the population in the long run [49]. Furthermore, Kula et al. has investigated the effect of the high Mn content in the food on the development of *Lymantria dispar* caterpillars [50]. The result of the study indicated that the first-instar mortality increased, period of development prolonged and food consumption expanded. Interestingly, the defensive mechanism of caterpillars against the surplus Mn in food was to translocate this element into frass and exuviae. 

For plants, excessive Mn is a major abiotic stress in plant agriculture worldwide [51]. It not only decreases photosynthetic rate, which by means of controlling the biosynthesis of photosynthetic pigments [52] and regulating stomatal conductance plus transpiration rate but leads to a decline in production and quality in crops [51]. As a toxic metal, Mn in excess can poison plants by generating reactive oxygen species (ROS) and triggering oxidative stress. If ROS are not well scavenged, it could cause lipid peroxidation and damage photosynthetic pigments [53]. 

For microorganisms, Mn pollution mainly affects the function, principal component, variation of typical variables, stability and diversity of microbial community. According to a previous study, soil microbial community principal component analysis showed that there were significant differences in microbial communities between Mn-contaminated soil and normal soil (*p* < 0.01). When the concentration of soil Mn^2+^ was greater than 300 mg/kg, the variation of typical variables (e.g., dispersion) increased and the stability of the soil microbial community decreased. Microbial community diversity analysis showed that Mn-contaminated soil played an important role in McIntosh uniformity index of soil microbes (*p* < 0.01). The Shannon richness index showed that soil microbial richness was promoted when soil Mn^2+^ concentration was applied in the range of 250–350 mg/kg [54].

### 2.4. Transporters of Mn in Living Organisms

Strict homeostatic control is strongly required in case of Mn deficiency or Mn overload. At the systemic level, this kind of control is maintained mainly by the intestine and the liver [47]. The intestine regulates Mn absorption from daily diets, whereas the liver clears Mn from the blood and secretes it as a bile conjugate for subsequent intestinal reabsorption or fecal excretion [55]. By at least three decades’ research, Mn is transported via the divalent metal transporter 1 (DMT1) [56], the transferrin receptor (TfR) [57] that mediates trivalent Fe and Mn uptake, the divalent ion/HCO_3_^−^ ion symporters ZIP8 and ZIP14 [58], the solute carrier-39 (SLC39) family of zinc transporters [59], park9/ATP13A2 [60], the magnesium transporter HIP14 [61] and the transient receptor potential mela statin 7 (TRPM7) channels/transporters [62]. 

Several proteins that may transport Mn have been identified in plants, such as members of the CAX, CDF, or P2A-type ATPase families, which are all involved in intracellular Mn traffic. *Arabidopsis* Natural Resistance-Associated Macrophage Protein1 (NRAMP1), rice NRAMP5 and barley NRAMP5 are involved in Mn uptake into roots [63].

## 3. Abiotic Treatment of Mn

So far, several main physicochemical methods have been employed in Mn treatment, including chemical precipitation [64], adsorption [65], ion exchange [66], and electrochemical method [14]. Recently, one study has shown that activated sludge can effectively remove heavy metal in sewage, which functions as a cheap and efficient way to handle the problem of heavy metal contamination in wastewater [67]. Although these traditional approaches have their own excellent superiorities, their defects should be noticeable. For example, consuming high energy and generating by-products tend to inhibit the large-scale application of abiotic methods in treating Mn. Here is a table expanding on the advantages and disadvantages of different methods (Table 1).

## 4. Biological Treatment of Mn

Nowadays, bioremediation technologies based on microorganisms have attracted scientists’ great attention because of their remarkable advantages including high efficiency, low-cost, environmentally friendly and without secondary pollution [68]. Microbial treatment of pollution is cheaper than physical and chemical methods because of its small land area, relatively simple facilities and operations, low energy consumption, and no need to deal with secondary pollution. Microorganisms can overcome the stress from toxic heavy metal ions through rapid mutation and evolution by developing resistance systems. In addition, with the advantages of rapid reproduction, strong metabolic capacity and considerable types of species, microorganisms can effectively reduce the concentration and toxicity of heavy metals in the environment through a variety of mechanisms.

Here, typical bioremediation methods by microorganisms for heavy metal Mn pollution have been summarized comprehensively, focusing on biosorption, bioaccumulation, biological oxidation and MICP. All of them are important remediation pathways and have been widely applied to environmental treatment [69]. Some typical examples are also presented in Table 2. Comparatively, biological oxidation can endure lower Mn^2+^ concentration but attain higher removal rate while biosorption behaved inversely. Most reactions happened during the range of pH of 5–7.

### 4.1. Biosorption

#### 4.1.1. Mechanisms

Biosorption refers to a process of sorption/composition of soluble metals by microbial biomass’ chemical activity or the use of materials from bio-based sources [79]. The mechanism of cell surface adsorption is that the functional groups of the cell wall -COOH, -NH_2_, -OH, etc., are combined with metal ions or coordinate with each other in other ways [78].

#### 4.1.2. Influencing Factors

There are several factors influencing the effect of Mn removal, including contact time for biosorption, biomass concentration, pH, temperature and Mn concentration, as shown in Table 2 [78,80]. The biosorption of Mn increased with time and microorganisms had a certain biosorption time for taking maximum Mn. For example, the time taken for maximum Mn biosorption of 19.34 mg/g by *Aspergillus niger* and 18.95 mg/g by *Saccharomyces cerevisiae* was 60 and 20 min, respectively. The biosorption of Mn increased with rise in biomass and Mn concentration [81]. Mn uptake increased gradually with rise in initial pH from 2 to 7. For example, biosorption capacity of *Pseudomonas aeruginosa* AT18 for Mn^2+^ in water increased with rising pH in the pH range of 5.46–7.72 [82]. The process that the abundant H^+^ ions in the solution competes against Mn ions for attachment to binding sites of the biomass may explain the reason of lower Mn uptake at an acidic pH of 2. In contrast, at alkaline pH (at pH 9 or 11), the Mn uptake was negligible, which is reasonable, as metals tend to exist in the form of hydroxide colloids (precipitates) at alkaline pH, causing lower rate of biosorption [83]. Therefore, the acidic condition is beneficial for biosorption. It is reported that the biosorption capacity of the *Bacillus cereus* strain HM-5 (a kind of bacteria) for the Mn ions reached up to 98.9% at 600 mg/L initial metal ion concentration [78]. Studies involving the absorption of *Bacillus*, *Saccharomyces cerevisiae*, and *Pseudomonas* sp. 4−05 have embodied high efficiency of removing Mn, specifically, the removal rate up to 96% [84].

#### 4.1.3. Advantages and Disadvantages

Biosorption offers several advantages over conventional treatment methods, such as cost effectiveness and efficiency. Moreover, it is capable of minimizing chemical/biological sludge, and can regenerate itself with the possibility of metal recovery [85]. Nowadays, the routine of low-cost adsorbents derived from plant or agricultural by-products as a substitute for expensive traditional removal methods of heavy metal from discarded streams has been investigated [86,87]. The transformation in the source of adsorbents profoundly enhances the recycling of resources.

### 4.2. Bioaccumulation 

#### 4.2.1. Mechanisms

Some confusion has prevailed in certain literature regarding the use of the terms “bioaccumulation” and “biosorption” considering the state of the biomass. Ozdemir et al. defined bioaccumulation as a phenomenon in living cells, whereas biosorption mechanisms are based on the use of dead biomass [88]. Heavy metal biosorption by living cells is often composed of the rapid initial surface binding followed by a second, slower phase of transport across the plasma membrane into the cell [89]. This second phrase is named “bioaccumulation”, which represents the concomitance of adsorptive and metabolism-dependent mechanisms, and contrasts “biosorption” because it does not involve metabolic contribution and can also be affected by non-viable biomass [90].

#### 4.2.2. Influencing Factors

Primary influencing factors are microorganisms’ periods of growth phase and pH. Meanwhile, there is a special mechanism remaining to be affirmed.

It is claimed that several species of macro fungi are capable of accumulating heavy metals in the different stages of the development of the organism, namely mycelium, sporophore and rhizomorphs, if present [91]. Similarly, Yilmaz et al. also revealed that the bioaccumulation capacity of Mn^2+^, Zn^2+^, Cu^2+^, Ni^2+^, and Co^2+^ by *Bacillus circulans* strain EB1 indicated variation in different periods of growth phases [92]. Ozdemir et al. reported that the highest bioaccumulation capacity of Mn^2+^ during 24 h incubation performed by *Geobacillus thermantarcticus* and *Anoxybacillus amylolyticus* was 24.5 (8 h) and 28.5 (20 h) mg/g dry weight, respectively, and determined that the highest metal capacity which was bioaccumulated by both bacteria was Mn, compared with Cd^2+^, Co^2+^ and Cu^2+^ [93]. This study also discovered that there was variation in different periods of the growth phases in terms of metal bioaccumulation capacity. Furthermore, generally, it was at the end of the stationary phase (20 h) that the maximum capacity occurred. Other researchers confirmed that an acidic surrounding (pH < 5) where there are many protons like H_3_O^+^ and H^+^ may reduce the Mn^2+^ adsorption rate because it is difficult for Mn^2+^ to bond with the cell walls [94]. Noszczynska et al. isolated and determined a *Pseudomonas* sp. strain, which exhibited huge Mn removal potential from metallurgical waste heap. They supposed that their cells have special mechanisms to store Mn considering that they show high capability and specific process of accumulation [76].

### 4.3. Bio-Oxidation

#### 4.3.1. Mechanisms

Biological oxidation of Mn means Mn-oxidizing bacteria (MnOB) oxidize the divalent Mn ions dissolved in water to trivalent or tetravalent Mn ions, and then form either Mn^3+^ or Mn^4+^-oxi-hydroxides (Mn bio-oxidation), so as to achieve the effect of removing the heavy metal Mn [95]. Microorganism-mediated oxidation of Mn is several magnitude orders faster than non-biological Mn oxidation [96]. Biological oxidation is one of the most feasible methods to treat Mn in the waste in mineral water, groundwater and artificial sewage. 

MnOB are most likely isolated from certain places, including deep sea nodules, Mn-rich biofilms, freshwater lake sediments, ferro Mn deposits, submarine basalt surfaces [97] and hydrothermal vents [98], which are essentially important in Mn bio-oxidation. Naturally, the process of Mn-oxidation without MnOB happens at a very low speed with the solution pH ranging from 6.0 to 9.0 [99]. 

The ability to oxidize Mn has been observed in a diverse group of bacteria, such as Bacillus sp. strain SG-1, *Leptothrix discophora* strain SS-1 and *Pseudomonas putida* strains MnB1 and GB-1. *Leptothrix* bacteria are known for their sheath-like structure and their ability to oxidize Mn, and the species *Leptothrix discophora* SP-6 has been studied extensively as a model organism [100,101,102]. For example, the removal efficiency of Mn was up to 90% as long as a *Leptothrix* strain were inoculated in the filtration columns [103]. In addition, MnOB showed distinct morphologies of their colonies when grown on plates containing Mn, while they formed normal colonies in the absence of Mn, which may indicate that the morphologically distinct structures produced by the bacterial colonies assist these bacteria to perform this function of Mn-oxidation [104]. 

Mn bio-oxidation occurs in two different pathways, but some experts believe there are chances that they happen simultaneously [105,106,107]: (i) Direct bio-oxidation, which is mediated by specific enzymes in the cells [69,108,109], and (ii) Indirect bio-oxidation, depending greatly on the pH and redox conditions of the environment which results from bacterial metabolites and/or microbial growth [105,110].

Direct oxidation relies on MnOB to facilitate the oxidation of Mn with the aid of a specific Mn oxidase to fasten the process of oxidation, or through other macromolecular substances such as proteins and polysaccharides in cell membranes that collect, adsorb and bind Mn [111]. So far, the key enzymes in biological oxidation of Mn are predominantly multicopper oxidases (MCOs), lactase, Mn peroxidase, and lignin degradation enzymes [112,113,114]. Among them, MCOs are more widely discussed. MCOs refers to a family of enzymes, utilizing copper as a cofactor in coupling the oxidation with the reduction of O_2_ [113]. Specific processes include using oxygen molecular as the oxidant, transferring electrons from the substrate to molecular oxygen via multicopper atoms existing in the enzyme. 

Previous research has demonstrated that several MCOs such as MnxG, MofA and MoxA can function as putative Mn oxidases [107]. MCOs have been found in a wide range of organisms including bacteria, fungi (laccase), plants, insects and vertebrates (ceruloplasmin) [115]. The oxidase *CueO*, a kind of MCOs, was identified as a laccase homologue. According to the phenomenon that the activity of *CueO* strongly improves in the presence of excessive copper (II) ions, *CueO* holds the ability of protecting cells against copper, even the mechanism is still unclear. In spite of their various functions, MCOs are similar in their structures, and most of them work through binding four copper atoms to two highly conserved copper centers [116]. A typical multicopper oxidase is made up of three cupredoxin-like domains. Highly conserved residues in domains I and III take responsibility for coordinating the four copper ions, and those in domains II and III take charge of forming the substrate binding pocket [117,118]. The substrate specificity of MCOs is predominantly influenced by the size, shape plus specific residues of the substrate binding pocket, as well as the difference in redox potential between the T1 copper and the substrate [119,120]. Furthermore, the products of direct oxidation, biogenic Mn oxides, which closely surround the cells of the bacterium, can also act as a strong adsorbent of multiple heavy metal ions via electrostatic attraction, ion exchange or surface complexation [121].

The fact that MCOs type enzymes are responsible for Mn oxidation has been proved, whereas the specific mechanisms have not been fully illustrated. MCOs participate in pigment formation, siderophore oxidation, and bio-polymerization. Furthermore, the non-Mn-oxidizing phenotype is often caused by the disappearance of MCOs [94]. However, although some *Bacillus* strains possessed Mn oxidation activity, they still did not produce the MCOs [101]. The phenomenon indicated that Mn oxidase was not the single mechanism in all MnOB strains. 

Despite the fact that some bacterial species have genes encoding for MCOs and peroxidases, these enzymes are not always applied to Mn direct oxidation. They tend to proceed indirect oxidation, which relies more on the metabolic response of the oxidative bacteria to regulate the oxidation mechanism. It is certified that changes in pH, dissolved oxygen, metabolites have a significant effect on the oxidation process. During exponential growth phase of bacteria, they take advantage of the amino acids which are abundant in peptone and yeast extract in the culture medium to produce ammonium ions (NH_4_^+^), which leads to the increase of pH. During the stationary phase, apart from a pH increase, extracellular super oxides also play a significant role in indirect oxidation. [122] The oxidation of Mn^2+^ by the reactive oxygen species (ROS) superoxide (O_2_^−^) is thermodynamically favorable in all relevant pH conditions [123]. So, reaction 1 is easy to occur. However, the product hydrogen peroxide (H_2_O_2_) has a tendency to reduce Mn^3+^ back to Mn^2+^ ions (reaction 2). Fortunately, the catalase enzyme is able to catalyze the decomposition of H_2_O_2_ into H_2_O and O_2_ (reaction 3), weakening the process of reaction 2.
Mn^2+^ + O_2_^−^ + 2H^+^ → Mn^3+^ + H_2_O_2_(1)
Mn^3+^ + 1/2 H_2_O_2_ → Mn^2+^ +1/2 O_2_ + H^+^(2)
H_2_O_2_ → 1/2 O_2_ + H_2_O(3)

#### 4.3.2. Influencing Factors

Factors influencing the process of oxidation consist of pH value as well as concentrations of oxygen and cells. In general, under neutral or slightly alkaline conditions (around pH 6.5–8.5), the activity of Mn oxidase maintains at a high level [96]. Accordingly, strong acid and strong alkaline environments can suppress Mn oxidative activity [124,125]. However, this does not mean that acidic environments cannot admit of the existence of MnOB. A number of bacterial strains have been identified in extreme surroundings, such as *Bacillus altitudinis*, *Frondihabitans*, and *Sphingomonas* [126].

Another factor, the oxygen concentration (often expressed as percent saturation at 20 °C) in cultures in the late logarithmic phase, is relevant with the shaking rate during the cultivation, and it is directly proportional to the shaking rate in the range of 50 to 100 strokes/min using a reciprocating shaker [127]. Considering the impact of dissolved oxygen (DO) on the enzyme-mediated oxidation, Katsoyiannis and Zoubouli found that the biological removal of Mn requires more stringent DO conditions than that of iron in the two-stage up flow fixed-bed filtration unit [102]. The specific data is that removing Mn needs DO 3.8 mg/L which was obtained by the process of aeration from initial concentrations of 0.9 mg/L, while removing iron needs 2 mg/L. Similar effect of DO has also been identified in terms of *Pseudomonas fluorescens* GB-1 [127]. To be specific, Mn-oxidizing activity was barely detected as long as the oxygen concentration was under 14% saturation in the late exponential phase. When the DO concentration raised from 15% to 26%, the activity increased. Once the concentration was higher, the activity decreased proportionally. It is noted that the oxygen concentration in cultures during the early and mid-logarithmic growth phases tended to be close to zero, even at high shaking rates, which was probably on account of the bacteria’s high metabolic activity during these stages. 

Cell concentration is related to vital activity and metabolic level. To determine the effect of cell concentration on the rate of Mn oxidation, a series of Mn oxidation experiments on *Leptothrix discophora* strain SS-1 were conducted at four different cell concentrations ranging from 24 to 35 mg/L. The results demonstrated that the Mn oxidation rate was directly proportional to cell concentration, with a maximum specific oxidation rate of 0.0052 μmol Mn^2+^/(min · mg cell) [72].

#### 4.3.3. Advantages and Limitations

Removing Mn through MnOB is a prospective method for treating wastewater under neutral conditions, because after Mn^2+^ is oxidized to Mn^3+^ and Mn^4+^, the resultant oxide can further adsorb dissolved Mn^2+^ [108]. Although certain exploration concerning the biological oxidation of Mn under artificial wastewater has been carried out, and several MnOB have been discovered and isolated from several harsh environments, the technology of using MnOB to treat Mn mine wastewater biologically has not been completely mature. The reasons are as follows: (1) Strong acidic effluents of Mn ore and its higher Mn^2+^ concentrations result in great toxicity to MnOB; (2) Compared with acidic soil, nutritional ingredients in Mn ore effluents are so inadequate that constrain the growth and biological activity of MnOB. (3) In the meantime, for processing Mn ore, there are chances that coexisting ferrous ions coming from pyrite dissolution are discharged into wastewater and consume large amounts of O_2_ in the system, leading to insufficient DO concentration to maintain effective biological oxidation. Despite the fact that adding alkali lime or aeration can adjust the acid environment and increase the concentration of DO, these procedures are costly and probably cause secondary pollution [128]. Additionally, more understandings of MnOB on Mn homeostasis and resistance mechanisms can be explored by novel biomolecular technologies, such as the transcriptome analysis [129].

### 4.4. Microbially Induced-Carbonate Precipitation (MICP)

#### 4.4.1. Mechanisms

Another promising and eco-friendly technique for Mn immobilization is biomineralization by producing carbonate precipitation due to well stability of Mn carbonate [130]. Microbially induced carbonate precipitation (MICP) refers to forming carbonate in the presence of different metabolic activities of microorganisms, including urea hydrolysis, photosynthesis, denitrification, ammonification, sulfate reduction and methane oxidation [131]. Especially, MICP driven by urea hydrolysis is widely researched and has been applied to repair cracks of constructions and roads, solidify sand and gravels, manufacture self-healing cement, develop bio-mineralization materials and so on, which contributes to the capability of bioprecipitates to improve the structure, increasing compressibility and decreasing the porosity and the permeability [132].

The process of MICP is dependent on ureolytic bacteria that hydrolyze urea through producing urease so that carbonate and ammonium ions are generated. The existence of ammonium leads to the increase of pH in the medium, which is favorable for carbonate to bond with divalent heavy metal ions, such as Cu^2+^ [133], Cd^2+^ [134], Pb^2+^ [135] and eventually promote the formation of carbonate precipitates. For Mn, the dominating products are rhodochrosite (MnCO_3_) precipitation [136] or CaMn (CO_3_)_2_ via substituting for Ca^2+^ in the crystal lattice of calcite, thus converting Mn from the soluble to insoluble mineral forms [137]. Equations (4)–(10) show the biochemical reactions in the process of MICP driven by urea hydrolysis [138].
CO(NH_2_)_2_ + H_2_O → NH_2_COOH + NH_3_(4)
NH_2_COOH + H_2_O → NH_3_ + H_2_CO_3_(5)
H_2_CO_3_ ↔ HCO_3_^−^ + H^+^(6)
2NH_3_ + 2H_2_O ↔ 2NH_4_ ^+^ +2OH^−^(7)
HCO_3_^−^ + H_+_ + 2NH_4_^+^ + 2OH^−^ ↔ CO_3_^2−^ + 2NH_4_^+^+ 2H_2_O(8)
Cell + Ca^2+^ → Cell − Ca^2+^(9)
Cell − Ca^2+^ + CO_3_^2−^→Cell − CaCO_3_(10)

In addition, Mn carbonate precipitation induced by ammonification pathway has also been investigated [139]. Bacterial activity can lead to the production of CO_2_ and ammonia via metabolism of amino acids [131]. Subsequently, ammonia hydrolysis would generate ammonium and OH^−^, leading to the pH increase. This alkaline environment facilitates the conversion of dissolved CO_2_ to carbonate ions. Once the solution conditions are sufficiently supersaturated, the precipitation of Mn carbonate would occur.

Four main biological mechanisms employed in Mn removal are shown in Figure 2. 

#### 4.4.2. Influencing Factors

Carbonate deposition with microorganisms is generally considered to be controlled by induction mechanism because the mineral type depends largely on environmental factors. The formation of carbonate in different environments is affected by microbial species and other abiotic factors, among which bacterial species, temperature, pH, concentrations of substrates and Mn as well as cultivation method that leads to distinct characteristics of precipitated crystals (size, morphology, microtopography, etc.) and sequestration efficiency have been acknowledged as primary and indispensable [138,139]. For example, the mean particle size of Mn carbonate crystals produced by *Ensifer adhaerens* using the streak plate method was twice as high as the particle size in the spread plate method, that is 14.9, 6.7 μm, respectively, which may contribute to early space restriction for the colony expansion in spread plate method [139]. The precipitation of rhodochrosite induced by *Sporosarcina luteola* were spheroidal concretions of <3 μm in size [136]. In the bio-modification of steel slag using MICP treatment, the soluble Mn is efficiently transformed into insoluble carbonate, and thus greatly reduced the leachability of Mn. The pH value is a comprehensive index about microbial metabolic activity in certain environment conditions, which has an effect on the growth of bacteria and product synthesis by controlling the microbial cell structure and the speed of using medium. Furthermore, pH has an important influence on the stability of Mn carbonate minerals, and near neutral to alkaline conditions are preferred [142]. However, due to limited research in this field, specific data in regulatory factors and molecular mechanisms have yet to be acquired, and further studies are needed to enhance our understanding and advance the actual application of MICP in Mn removal.

#### 4.4.3. Advantages and Limitations

When it comes to the advantages of MICP, simple operation and management, low processing cost, less secondary pollution and little interference to the surrounding environment stand out [138]. Thus, it is a kind of economic, effective, and non-destructive remediation technology. The resulting product Mn carbonate at nano- or micro-size has various applications as electrochemical devices, catalysts, and even adsorbents to remove other heavy metals from the environment, thus the secondary recycling and reuse of Mn are realized.

However, there is still a lot of room for further exploration and improvement. For example, the application of MICP may be restricted to engineering and sand embankments, and even if there are research to apply MICP to the heavy metal remediation, Mn is barely mentioned. In addition, specific studies on the influencing factors and characteristic of sediments in Mn treatments by MICP have not been carried out.

In fact, in the complex biological processes for Mn removal, different removal mechanisms mentioned above often occur simultaneously, sometimes containing abiotic precipitates. The removal of Mn^2+^ by *Aeromonas hydrophila* depended primarily on Mn-oxidation (49.55%) by either enzymatic catalysis or abiotic reactions, as well as biosorption by binding the negatively charged carboxylate groups of phospholipids and lipopolysaccharides on the cell wall [143]. For *Stenotrophomonas* sp., distinct mechanisms were observed at different Mn^2+^ concentrations, resulting from changes in intracellular metabolism. Mn^2+^ was principally fixed in Mn-oxides by bio-oxidization and MnCO_3_ via ammonification at 50 mM and 10 mM Mn^2+^, respectively [144]. Therefore, the bioremediation of Mn pollution in actual application still has space for improvement by customized regulation and optimization of various external conditions, unleashing the full potential of multiple removal mechanisms.

## 5. Combination Methods to Treat Mn

Sometimes, a single method is difficult to handle complex environments in the process of heavy metal treatment, so that fails to achieve ideal removal efficiency. Thus, scientists are creating effective combinations of different approaches to maximize the role of every single method. 

### 5.1. MnOB and Microalgae

Detailed information about MnOB has been specifically introduced in Section 4.3 “Bio-oxidation”. As far as microalgae, they are either unicellular or multicellular prokaryotic cyanobacteria and eukaryotic microorganisms that can experience photosynthesis and survive in tough environments, like the taxa *Cryptophyta*, *Chlorophyta*, *Euglenophyta Chlorarachniophyta*, *Rhodophyta*. Microalgae are capable of inducing ROS and releasing superoxide dismutase plus peroxidase to accommodate the high metal concentrations [145]. Therefore, microalgae have good resistance to harsh surroundings. Stuetz et al. have found that with the assistance of unicellular microalgae, the rate of bacterial Mn oxidation was increased by a factor of 10 than that proceed by bacterial oxidation alone [146]. 

Combining MnOB with microalgae simultaneously might reduce or avoid competition for nutriments, following the toxicity reduction of wastewater. By virtue of photosynthesis, both pH values and concentrations of DO in the media would increase, which not only provides suitable conditions for MnOB growth, but accelerates the oxidation of Mn^2+^ to Mn^4+^. It deserves to be mentioned that macroalgae are capable of accumulating Mn^2+^ through binding groups on cell surfaces in indirect oxidation [145]. In addition, microalgae can accelerate the rate of bacterial growth with the aid of secreted extracellular polymeric substances (EPS). Therefore, constructing a co-immobilized MnOB/microalgae system with high efficiency of Mn removal may be a pivotal step of successfully treating mine wastewater containing Mn^2+^ using cooperative biological methods [128]. However, the related mechanisms have yet to be verified and the corresponding process remain to be optimized. 

### 5.2. MnOB and BioMnOx

Biogenic manganese oxides (BioMnOx) produced by microorganisms have been considered as a promising material to eliminate toxic heavy metals and organics from polluted water and soils [147]. The desirable performance of BioMnOx on the absorption of heavy metals is ascribed to its large surface-to-volume porous structure, known as layered nanoparticle minerals [148]. Some researchers recently found that BioMnOx are helpful for the synergistic removal of thallium(I) [149]. In some cases, the adsorption sequences for different metals are generally distinct due to their different interaction natures with BioMnOx [148]. Apart from powerful adsorption capacity, BioMnOx is identified as one of the strongest oxidizing agents in the environment and show higher reactivity to interact or oxidize metal ions than abiotic Mn oxides [150].

A creative strategy of combining a kind of bacterium with BioMnOx generated by itself was proposed. With the help of BioMnOx, a novel highly Mn-tolerant bacteria, *Providencia* sp. LLDRA6, can immensely facilitate the removal of various heavy metals (Pb, Cr, Cd, Cu, Mn, and Zn) from contaminated soils, showing a higher removal efficiency than the bacterium alone [151].

### 5.3. MnOB and Biochar

In bioremediation, microbial immobilization has aroused great attention till now because it can, to a great extent, maintain the biomass of microbes in unfavorable environments [152]. Common immobilization carriers mainly consist of natural polymers, such as alginate, agar and chitosan, and synthetic compounds including polyacrylamide, polyurethane, polyvinyl alcohol and so on [153]. However, these carriers have certain shortcomings of low mechanical strength and high prices, limiting their large-scale applications. In contrast, biochar made from agricultural waste is renowned for low energy supply, convenient access to raw materials and simple preparation process. 

A method of combining *Acinetobacter* sp. AL-6, a kind of Mn-oxidizing bacterium [154] with pomelo peel biochar, exhibiting outstanding adsorption performance [155], was proposed to remove Mn together (Figure 3). Studies have shown that the microbe-biochar system has a tremendous synergistic effect on removing Mn. The toxic effect of Mn on strain AL-6 seems to be weakened by biochar. In addition, there are many different depths ravines on the surface of biochar, which could provide adequate adsorption sites for bacteria to maintain biological activity and ensure normal growth. Meanwhile, the bacteria are not likely to interfere with the adsorption sites of biochar [156]. The Mn removal rate in 48 h can reach up to 95% and can be maintained at 96.96%–98.48% [157]. Youngwilai et al. has made a similar attempt to combine Mn-oxidizing bacteria, *Streptomyces violarus* strain SBP1 with biochar [156]. The biochar was modified by H_2_O_2_, which enables it to possess a higher proportion of oxygen-containing functional groups by oxidizing its carbon surfaces, leading to more metal adsorption sites. The results showed that the Mn ion was bound on the biochar surface through bulk diffusion, leading to rapid adsorption in the initial period. After that, the metal ion diffused into inner micro-pores of sorbent and the highest removal efficiency was at 78% [158]. 

### 5.4. Microorganisms and Phytoremediation Plants 

Phytoremediation is an in-situ remediation technique that utilizes the root of plants to transfer, remove, and stabilize the contaminants in soils and sediments [159]. As candidate plants of phytoremediation, they are supposed to have characteristics of rapid growth, high biomass and strong capacity to tolerate high concentrations of heavy metals. At present, researchers have primarily focused on herbaceous plants when it comes to the phytoremediation in Mn-contaminated soil, including *Xanthium strumarium* and *Phytolacca americana*, but arbor plants such as *Pinus massoniana* have also been involved [160]. Specifically, *Broussonetia papyrifera* has been chosen as the pioneer species, not only because they exhibit preponderances mentioned above, but they have strong adaptability, show wide distribution and easy reproduction and to a great extent, have hyper resistibility to drought, salt and alkali stress environment [161,162]. *Bacillus cereus* HM5 and *Bacillus thuringiensis* HM7 are two strains that are capable of dissolving phosphorus, produce indole acetic acid (IAA) plus iron carriers and alleviate Mn toxicity [163]. The combination of *Bacillus* spp. and *B*. *papyrifera* has great application prospects in treating heavy metal contamination in soil. By pot experiments, biomass of *B*. *papyrifera* increased a lot compared to the control group after inoculated with two strains, mainly due to the fact that hormones produced by microorganisms help maintain cell division and elongation, thus increasing the length of stems and biomass. At the same time, Fe^3+^, accumulated in plant roots but not used, has a chance to bind to the iron carriers that are produced by two strains of bacteria and be made the best use to promote plant growth. However, it is worth paying enough attention on the mechanism that bacteria follow to collaborate with *B*. *papyrifera*. The principal component analysis showed that two *Bacillus* strains prefer to promote plant root function maintenance and improve soil environments, rather than induce direct biological adsorption or oxidation of heavy metals [164]. In a bid to maximize the remedying effects of combination between microorganisms and phytoremediation plants, persistent chases for candidate plants and effective bacteria shall not be neglected.

## 6. Conclusions and Future Perspectives

Mn, as an essential trace element, plays a significant role in regulating the human body’s growth and development. However, excessive Mn concentrations because of rapid industrialization cause notable damage to ecosystem and toxic effects on living beings, which has been a global concern. Bioremediation based on microorganisms holds a promising treatment potential for heavy metal contamination due to its high removal capacity, low-cost and eco-friendly characteristics. This review presents the main sources of Mn pollution, migration in biogeochemical cycle and a variety of adverse impacts on beings. A wide range of microbe-mediated approaches are introduced in detail, including biosorption, bioaccumulation, bio-oxidation and MICP, through adsorption and accumulation of microbial cells or transformation free Mn ions to insoluble mineral precipitations on account of microbial activities. Especially, the strategies and strengths of synergistic treatment for Mn via multiple pathways are summarized. Although these methods exhibit a variety of benefits and considerably trap Mn, there is a broad scope for future research, due to the complexity and variability of the natural environment.

In terms of biosorption research, more low-cost adsorbents derived from plants or agricultural by-products, functioning as economic substitutes for expensive traditional removal methods of heavy metal have yet to be produced.As far as biological oxidation, more microorganisms with strong ability to oxidize Mn remain to be explored and discovered and the activity of available microorganisms needs efforts to be improved, by optimization of environmental conditions or other methods.Although MICP has exhibited excellent feasibility in treating Mn pollution by converting Mn into Mn carbonates and then filtrating them, some important parameters of this method, such as influencing factors, characteristics of products, component analysis and application of products has not been clearly clarified. By referring to the more mature applications of MICP in other heavy metals removal, Mn remediation via MICP should transit gradually from laboratory level to few fields’ application in actual contaminated sites.Furthermore, more synergistic treatments of Mn involving various physical, chemical, biological mechanisms in a gesture to acquire higher efficiency and lower costs and overcome the intrinsic restrictions of single method could be explored extensively.

In a nutshell, only if the biochemical and molecular mechanisms of different types of microbial remediation are clarified, can researchers tap its full potential for removing Mn pollution. A comprehensive understanding of the relationships between Mn and other heavy metals or organic compounds in the environment is conducive to realizing simultaneous removal of them. Microorganism-mediated remediation of heavy metal is receiving more attention and has expansive prospect of application.

## Figures and Tables

**Figure 1 microorganisms-10-02411-f001:**
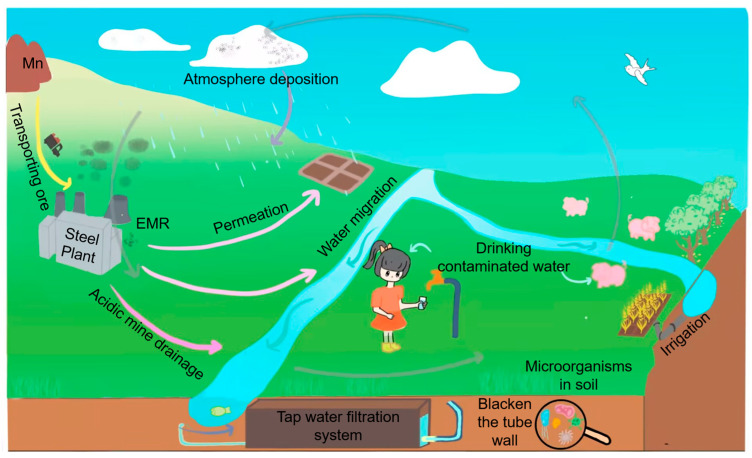
Sources, transfer processes and toxicity of Mn in the biogeochemical cycles.

**Figure 2 microorganisms-10-02411-f002:**
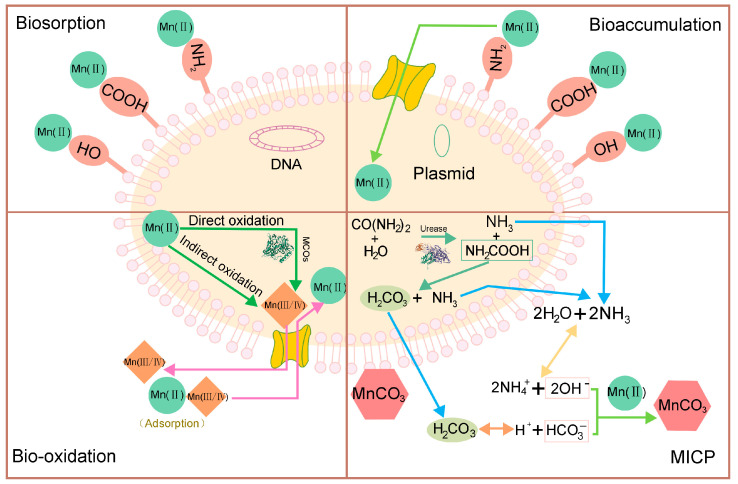
Illustration of four main biological treatments for Mn removal (where three−dimensional structures of a kind of MCOs and urease are presented). Reprinted with permission from Refs. [140,141].

**Figure 3 microorganisms-10-02411-f003:**
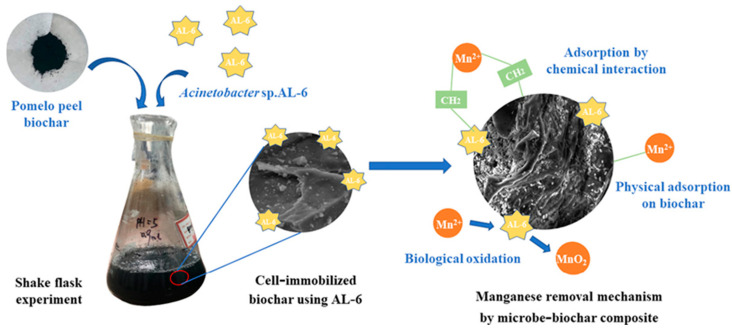
Mechanisms of synergies between *Acinetobacter* sp. AL-6 and pomelo peel biochar in a hybrid process. Reprinted with permission from Ref. [158].

**Table 1 microorganisms-10-02411-t001:** Advantages and disadvantages of different physicochemical methods for Mn removal.

Methods	Advantages	Disadvantages
Electrochemical method	Accurate regulation of kinetics; High removal efficiency	High operating costs; Large power supply
Ion exchange method	Non-toxic and renewable; High removal efficiency; Meeting the needs of industrialization	Ion exchange materials being susceptible to organic contamination in wastewater
Adsorption method	Easy access to adsorbents, such as activated carbon, zeolite, etc. Adsorbents being regenerative	Great tendency to cause adsorbent sludge, resulting in secondary pollution
Chemical precipitation method	Simple operation; Low cost; High removal efficiency	Easy to cause secondary pollution; Accumulation of a large amount of sludge

**Table 2 microorganisms-10-02411-t002:** Common bacteria for Mn removal via various mechanisms, including bio-oxidation, bioaccumulation and adsorption at their optimal conditions.

Mechanisms	Species	Initial Mn^2+^ Concentration (mg/L)	Optimum Temperature	Optimum pH	Removal Efficiency	Experimental Time	Refs
Biological oxidation	*Lysinibacillus* sp.	54.94	37 °C	7.0	94.7%	7 days	[70]
Biological oxidation	*Bacillus* sp.	1.65	24 °C	7.5	>83.3%	-	[71]
Biological oxidation	*Leptothrix discophora*	4.47	30 °C	7.5	97.5%	3.5 days	[72]
Biological oxidation	*Citrobacter* sp.	53.0	27 °C	7.0	76.2%	4 days	[73]
Biological oxidation	*Acinetobacter* sp.	200	Not mentioned	Self-regulation	99.1%	6 days	[74]
Bioaccumulation	*Papiliotrema huenov*	110	30 °C	5	75.6%	5 days	[75]
Bioaccumulation	*Pseudomonas* sp.	43.5	28 °C	Self-regulation	88%	18 days	[76]
Biosorption	*Serratia* sp.	500	34 °C	6.0	96.8%	76 h	[77]
Biosorption	*Bacillus cereus*	600	35 °C	6.0	60.3%	5 days	[78]

## Data Availability

Not applicable.

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
