# Peer review of "Manganese Pollution and Its Remediation: A Review of Biological Removal and Promising Combination Strategies"

_microorganisms, 2022, doi:10.3390/microorganisms10122411_

Round 1
Reviewer 1 Report
Mn bioremediation is an exciting and booming area of exploration. The authors have done an excellent job in compiling literature on biological removal strategies with Mn and various combination strategies. The text is well-referenced and coherent. From my perspective, the conclusion and viewpoints shed light on developing comprehensive techniques for Mn re-mediation in practical application.
Since Mn removal is a topic of interest to the researchers in certain fields, and information in this manuscript were found relatively updated and sufficient, I propose that it can be published after some subtly revisions are made. The suggestions are specific as follows:
1) Some microbial species in Table 2 picked up from different references are not in a canonical and consistent style. The Species “2De strain” , for example, should also be notated as “genus name + species name + strain name (if there is) ”. It is also appropriate to omit the word “strain”, but it must be consistent with the context. They should be carefully proofed in the full text.
2) This manuscript has cited many new literatures, but I still recommend that the author consult some latest valuable papers ( e.g. doi: 10.1016/j.jenvman.2022.116157; doi: 10.1016/j.scitotenv.2022.157394; doi:10.1016/j.scitotenv.2022.154865; doi: 10.1007/s12205-021-2263-3 and doi: 10.1016/j.jece.2022.108479)
3) The expression style of metal ion valence are also inconsistent in the paper. For instance, the representations of Mn2+ , Mn(â…¡) and Mn(â…¢) have been used.
4) The unit of concentration in the manuscript is not uniform (such as “mg L-1”and “mg/L” ). They should be carefully modified in the revision version.
5) Since the pollution, toxicity and treat methods of Mn were detailed introduced in the first three parts of the paper, I suggest the authors to modify the title as“Pollution and its remediation of manganese: a review of biological removal and promising combination remediation strategies”.
6) Table 2: Authors have introduced the concept of Biological treatment of Mn very well. Also, in Table 2, the authors have done a decent job compiling the research studies. However, as a reader, these case studies listed in this table have not been discussed very nicely in the texts. Hence, the authors might well reconsider a detailed explanation for Table 2 at relevant subsection to identify and compile these example studies.

Reviewer 2 Report
The manuscript with the title "Biological removal of Mn by microorganisms and promising combination remediation strategies: A review" presents an interesting contribution to the Mn remediation. In addition, the manuscript is well designed and easy to follow. The topics are adequate and the literature cited is actualized. The conclusion and future perspectives section is very concise and well discussed. It deserves publication in Microorganisms after minor revisions, according to the following comments:
11) Do not use the words in the title as your keywords. Change the keyword “Mn removal” and “Microorganisms”.
22) Why the synthesis process is considered a low-cost process? The "low-cost" approach is only in the title and is not justified in the manuscript. I suggest authors provide another title or address the reason in the manuscript.
33) Improve academic English throughout the manuscript. There are 45 "which" in the manuscript. Avoid overuse. Rewrite these passages.
44) The numbering of the equations should be revised. The authors started at 1 twice. Revise it.
Reviewer 3 Report
The subject area of the review is devoted to an extremely urgent problem of environmental pollution by emergent pollutants, manganese (Mn) in particular. The presented information is introduced in sufficient detail, clearly, and convincingly: ranging from the main sources of Mn pollution in open ecosystems, its migration in the biogeochemical cycle, and the adverse effects of Mn on living beings to the description of various biotechnological approaches to inactivation of this dangerous ecopollutant. At the same time, the authors used a fairly wide temporal coverage of literary sources—from 1984 to 2022.
In order to improve the manuscript, the authors are advised to consider the following comments:
(1) Abstract: it is recommended to give (Line 8) the abbreviation Mn in full as manganese (Mn). The same is applicable to page 28.
(2) In this review, the essentiality, toxicity and transporting of Mn are first introduced. There are works describing the toxicity of manganese and its transport systems and therefore it is advisable to refer to them (e.g., Bhattacharyya-Pakrasi M. et.al. Manganese transport and its regulation in bacteria (2002); Manganese in Health and Disease (2014); Bosma E.F. et al. Regulation and distinct physiological roles of manganese in bacteria. FEMS Microbiology Reviews, 2021, 45, 1–16) and then to rephrase this sentence.
(3) Give the full names of the abbreviations EMR (Line 111), LLDRA6 (Line 563) and IAA (Line 604).
(4) Section 5.4. Microorganisms and phytoremediation plants.
It is necessary to consider in more detail which plants can be manganese phytoremediation plants.
(5) Improvement is required in the design of the List of References. For example, Ref. 2: the year of publication is not in bold, but in further references it is in bold. Ref. 6: the number of the issue is not in italics. Somewhere, page numbers and DOIs are missing, etc.
I believe that the manuscript deserves to be published after these minor comments have been eliminated without the second review of the revised version.
